# Polygenic Risk Score Impact on Susceptibility to Age-Related Macular Degeneration in Polish Patients

**DOI:** 10.3390/jcm12010295

**Published:** 2022-12-30

**Authors:** Anna Wąsowska, Sławomir Teper, Ewa Matczyńska, Przemysław Łyszkiewicz, Adam Sendecki, Anna Machalińska, Edward Wylęgała, Anna Boguszewska-Chachulska

**Affiliations:** 1Chair and Clinical Department of Ophthalmology, Faculty of Medical Sciences in Zabrze, Medical University of Silesia, 40-752 Katowice, Poland; 2Genomed S.A., 02-971 Warszawa, Poland; 3First Department of Ophthalmology, Pomeranian Medical University, 70-204 Szczecin, Poland

**Keywords:** age-related macular degeneration, AMD genetics, polygenic model, polygenic risk score, susceptibility variants

## Abstract

Age-related macular degeneration (AMD) is a common retina degenerative disease with a complex genetic and environmental background. This study aimed to determine the polygenic risk score (PRS) stratification between the AMD case and control patients. The PRS model was established on the targeted sequencing data of a cohort of 471 patients diagnosed with AMD and 167 healthy controls without symptoms of retinal degeneration. The highest predictive value to the target dataset was achieved for a 22-variant model with a *p*-value lower than threshold P_T_ = 0.0123. The median PRS for cases was higher by 1.1 than for control samples (95% CI: (−1.19; −0.85)). The patients in the highest quantile had a significantly higher relative risk of developing AMD than those in the lowest reference quantile (OR = 35.13, 95% CI: (7.9; 156.1), *p* < 0.001). The diagnostic ability was investigated using ROC analysis with AUC = 0.76 (95% CI: (0.72; 0.80)). The polygenic susceptibility to AMD may be the starting point to expand AMD diagnostics based on rare highly penetrant variants and investigate associations with disease progression and treatment response in Polish patients in future studies.

## 1. Introduction

Age-related macular degeneration (AMD) is a common retina degenerative disease with a complex background, including genetic, environmental, demographic and geographical factors. The early symptom of AMD is the accumulation of protein-lipid deposits, called drusen, on the Bruchs membrane underneath the retinal pigment epithelium (RPE), disturbing its function and causing photoreceptor loss in the macula [1]. The progression results in late AMD classified as dry AMD with geographical atrophy (GA) or neovascular AMD with macular neovascularization (MNV). The disease constitutes the most common reason of severe vision impairment and blindness in developed countries [2]. It is estimated that approximately 67 million people currently suffer from AMD in the European Union and, as the Li et al. meta-analysis states, the number is about to increase by 10 million by 2050, as the population continues to age [3]. An effective treatment exists for neovascular AMD (intravitreal anti-VEGF therapy, IAI) [4] and its early administration when MNV is present, but before the appearance of symptoms, could be facilitated by the identification of genetic risk variants for AMD in the patient.

Due to the fact that the development of AMD may remain symptom-free for a long time, or because these symptoms do not attract the patient’s attention enough, a careful risk assessment could help determine the frequency of ophthalmic follow-up visits or identify the group of patients who benefit most from self-monitoring [5]. Currently, such self-monitoring is possible based on home optical coherence tomography, but the high cost makes it even more necessary to determine the optimal group to be screened. The genetic risk factors have been reported as playing a vital role in the development of AMD. Several studies, including those performed on the Polish population, identified two major coding variants, rs1061170 and rs10490924, in the *CFH* and *ARMS2* genes, respectively, with a strong impact on the risk of AMD, explaining at least 50% of the AMD heritability [6,7,8]. These common polymorphisms were also associated with the disease progression and wet type of AMD [9,10,11]. In addition, the large genome-wide association study performed by Fritsche et al. [1] identified 34 loci in the genome that mainly include genes for the complement system, lipid metabolism and extracellular matrix remodelling, which are involved in the AMD pathology. Fritsche reported 52 independently associated AMD susceptibility variants, from which four variants (rs2284665, rs10922109, rs116503776, rs2230199) were confirmed by Yan et al. [12] to also have an impact on the disease progression, indicating the shared genetic aetiology of the two traits of AMD.

As in most complex diseases, the cumulative risk is a derivative of contributions of numerous variants in many genes, with a low or modest individual impact on the disease risk. One of the methods to stratify one’s genetic predisposition for a disease, combining the weighted impact of multiple variants, is by calculating the Polygenic Risk Score (PRS). The PRS is estimated based on the effect sizes of risk alleles derived from GWAS summary statistics of a related trait with shared genetic aetiology and genotyping data of a target group of patients, such as a whole genome, exome or targeted sequencing or array data. PRS assessment has been successfully incorporated in several medical studies, for example in schizophrenia and breast cancer [13]. When combined with nongenetic factors, such as sex, weight or lifestyle, PRS may be a useful clinical tool to facilitate diagnosis, refine individual risk, perform population screening programs or even guide medical treatment [14,15].

## 2. Materials and Methods

### 2.1. Base Data

The base data comprised summary statistics of over 16 million variants determined by a GWA study of disease progression in age-related macular degeneration, publicly available on the NHGRI-EBI Catalog of human genome-wide association studies website [16]. The summary statistics concerned AMD occurring in 3685 cases together with 52,952 control samples from the GERA European ancestry cohort analysis. The additive inheritance model was used for calculations.

### 2.2. Study Group and Sequencing Data Analysis

The participants of the study were recruited by the Chair and Clinical Department of Ophthalmology, Faculty of Medical Science, Medical University of Silesia in Katowice and the First Department of Ophthalmology, Pomeranian Medical University in Szczecin. The project was approved by the Ethics Committee of the Medical University of Silesia (Resolution No KNW/0022/KB1/105/13) and adhered to the tenets of The Declaration of Helsinki. The participants were informed about the study’s purpose and the study protocol. Informed written consent was obtained from all of the participants. The cohort included 471 patients diagnosed with AMD and 167 healthy controls without any symptoms of retinal degeneration.

Patients of the above mentioned departments of ophthalmology (including outpatient clinic) were interviewed and initially examined during a routine appointment. Patients over 50 years of age were invited to the study. Patients with AMD were consecutively included in the study group during retinological follow-up visits at the ophthalmology clinic—both those with advanced forms of the disease (neovascular and geographic atrophy), as well as those with early and intermediate forms. The control group consisted mainly of people who were recruited during the follow-up visit after cataract surgery, in whom a thorough examination of the fundus did not confirm the signs of AMD. The following ophthalmological examinations were performed: best corrected visual acuity tested with the ETDRS charts. Pupillary dilation was achieved with 1% tropicamide (Polpharma, Starogard Gdański, Poland) after anterior segment slit lamp biomicroscopy (slit lamp Zeiss SL120, Carl Zeiss Meditec, Jena, Germany). A clinical examination of each eye was performed and was aimed at assessing the anterior and posterior segments of the eye (Volk superfield aspheric lenses 90D). The patients were subjected to further diagnostics to identify those with pathological changes in the macula. Digital images of the fundus, cross-section macular OCT (radial and 3D wide scanning protocols) and OCT angiography (OCT-A) with a 6x6 mm macular cube were taken with swept-source optical coherence tomography using the DRI OCT Triton tomograph (Topcon Healthcare, Tokyo, Japan). Moreover, extended clinical data, including a detailed description of the medical history, current medical treatment as well as physical parameters (e.g., body mass index), smoking status, and alcohol consumption were collected.

The targeted enrichment of coding sequences of 30 AMD associated genes (listed in Table 1) with flanking intronic regions using Molecular Inverted Probes and Illumina sequencing was performed in Genomed S.A., Warsaw, as described previously [8]. The bioinformatic analysis included adapter trimming using Cutadapt v1.14 [17], mapping reads to the GRCh37.13 reference genome with a Burrows-Wheeler Aligner v0.7.10 [18] and deduplication based on Unique Molecular Identifiers using in-house scripts. A Genome Analysis Tool Kit (GATK) v3.5 [19] was used for best practices indel realignment and base recalibration. Variant calling was performed with both HaplotypeCaller and UnifiedGenotyper from the GATK package in order to ensure the best SNP and indel identification. The resulting variants in gVCF were initially filtered, excluding variants missing in less than 95% of samples and low coverage variants with coverage lower than 10x in 80% of genotypes. 

### 2.3. Target Data QC

The set of 2348 variants were treated as the target data for QC and PRS scoring. The initial target data QC was performed using PLINK v1.90 [20], according to Marees et al. [21], as suggested by Choi et al. [22]. Individuals with genotype gaps > 5% and variants with genotyping rate < 5% were filtered out, as well as extremely rare SNPs with MAF < 0.3% (as used by Ulańczyk et al. [8]) and variants deviating from Hardy-Weinberg equilibrium law (*p* < 1 × 10^−10^ for cases and *p* < 1 × 10^−6^ in controls). Heterozygosity was evaluated by the F-coefficient estimation available in PLINK and samples within three standard deviations of the mean were retained. Ambiguous, mismatching and duplicated variants were verified using custom in-house R scripts. There was no sample overlap or relatedness between the base and target samples. Population stratification of pruned target data was checked using the data from 1000 Genomes, assuming the closest genetic distance to samples of European ancestry. Cohort outliers were checked using the Mahalonobis distance. The monogenic variant association to AMD was estimated using logistic regression analysis with sex, BMI, smoking status and the first four genetic distances from multi-dimensional scaling as covariates. 

### 2.4. PRS Calculation

To assess the Polygenic Risk Score, PRSice-2 [23] with an additive model used for regression was used. The analysis focused on variants that overlap between the base and target data. As the PRSice-2 utilizes the “Clumping + Thresholding” method, the clumping parameters were adjusted similarly to those used in the linkage-disequilibrium-based pruning procedure in the population stratification step of the target data QC performed using PLINK (window size set to 100 kb and squared correlation threshold r^2^ set to 0.5). Clumping retained the variant with the smallest *p*-value at the locus in linkage disequilibrium as the independent effect SNP. Calculating the best-fit PRS model included (i) the first three genetic multi-dimensional scaling components, (ii) sex, age and smoking status in terms of pack years, and (iii) both covariate sets combined. The variance explained, reported as Nagelkerke’s pseudo-R^2^, was calculated as the R^2^ of the full model (phenotype ~ PRS + covariates) minus the R^2^ of the null model (phenotype ~ covariates). In order to avoid an overfitted prediction model, the empirical *p*-value was calculated by performing 10,000 permutations. Additional statistical analysis and result visualisations were performed in R software v3.4.4 (The R Foundation for Statistical Computing, Vienna, Austria) using the plyr [24], genetics [25], pROC [26] and ggplot2 [27] libraries. The diagnostic ability of the model was calculated using Area under the Receivers Operator curve (AUC) representing the probability of a random positive result in the model prediction. 

## 3. Results

### 3.1. Study Group

Genotype and phenotype data from the NeuStemGen project consisted of 638 participants, out of which 471 were AMD cases and 167 were controls. The overall age at enrolment time ranged from 40 to 93 years, with mean (SD) 74.7 (7.9) and 72.1 (7.0), for cases and controls, respectively. Women represented 65.1% (*N* = 481) of all participants. The detailed characteristics of cases and control cohorts are presented in Table 2.

### 3.2. Target Data QC

A total of 2348 variants in the target data were intersected with variants available in base data summary statistics. Some 650 variants genotyped in 638 individuals were retained in the analysis and underwent quality control. One variant and two samples were removed due to missing genotype data. A total of 115 extremely rare SNPs with MAF < 0.3% were filtered out from the dataset. Ten variants did not meet Hardy-Weinberg Equilibrium criteria. Three samples were rejected because of deviating by more than three standard deviations from the mean heterozygosity rate. As expected, a population stratification analysis confirmed the closest proximity to samples of European origin. 

### 3.3. PRS Analysis

The intersection of base and target data resulted in 524 variants that passed the data QC. After clumping, 311 variants were retained in the polygenic risk score calculation. The highest predictive value to the target dataset was achieved for a 22-variant model with a *p*-value lower than threshold P_T_ = 0.0123. The inclusion of covariates changed neither the *p*-value threshold nor the permutation *p*-value, but affected the full model R^2^ and the proportion of variance explained by the PRS model (shown in Table 3). The goodness-of-fit of the estimated risk prediction model was confirmed using the Hosmer-Lemeshow test (χ^2^ = 9.9, *p* = 0.27).

The variants included in the PRS model are described in Table 4. Although the enrichment panel included regions from 30 genes (Table 1), variants in nine genes (*ABCA4, ARMS2, C2, C3, CFB, CFH, HTRA1, NELFE, TLR4*) exceeded the threshold *p*-value and were selected by the algorithm for PRS estimation. A total of 18 out of 22 variants were submitted to the dbSNP and ClinVar database and, to date, 14 of them had been reported to have an impact on macular degeneration (Table 4). 

The overall median PRS for cases was higher by 1.1 than for control samples (−0.57 vs. −1.68, 95% CI: (−1.19; −0.85)), as shown in Figure 1, and the statistical significance was confirmed using the Mann-Whitney test with *p* = 2.2 × 10^−16^. 

The statistical significance using the Mann-Whitney test with *p* < 0.001 was also proven in subsets when sex and the patient recruitment region (Katowice, Szczecin) were concerned (shown in Figure 2).

The disease risk increases with increasing PRS value, and patients in the highest quantile had a significantly higher relative risk of developing AMD than those in the lowest reference quantile (OR = 35.13, 95% CI: (7.9; 156.1), *p* < 0.001), as presented in Figure 3. The diagnostic ability was investigated using ROC analysis with AUC = 0.76 (95% CI: 0.72; 0.80). 

Taking age into consideration, the median PRS values for cases in the age groups 60–69, 70–79 and 80+ were significantly higher with *p* < 0.001. Further analysis showed that 92% of individuals with PRS in the 10th decile developed AMD (Figure 4).

## 4. Discussion

Heesterbeek et al. [28] gathered 25 AMD prediction models of late stage AMD with their discriminative performance and proved that the prediction power depends on the number and type of risk factors included, with only a few variants considered as genetic factors tested. However, the development of risk models should not be restricted to a low number of AMD-associated variants, as the effect size of many other variants discovered using GWAS may influence the genetic risk of the disease. We determined a 22-variant polygenic risk score predictive model based on targeted sequencing of AMD-related genes and GWAS studies on AMD. The model provided a differentiation of disease status in the cohort of 638 participants and confirmed the substantially higher risk of AMD development in patients with the highest PRS values. The established PRS model reached a medium discriminative value AUC = 0.76, comparable to values established in studies on other diseases (systemic sclerosis AUC = 0.673, inflammatory bowel disease AUC = 0.72, systemic lupus erythematosus AUC = 0.62–0.78, breast cancer AUC = 0.63) [29,30]. 


*ABCA4*


The intron variant rs2297634 has been reported as significant in the Age-Related Eye Disease Study (AREDS), but was not confirmed in a replication group of a non-Hispanic white population tested by Ryu et al. [31] using a logistic regression model. The variant does not show a splicing effect as analysed in the varSEAK (JSI medical systems GmbH, Etteheim, Germany) Online Splice Site Prediction tool (https://varseak.bio/index.php, accessed on 16 November 2022).


*ARMS2*


The rs10490924 in the *ARMS2* gene, causing alanine to serine amino acid change (A69S), is reported as the second best-known variant of AMD susceptibility [1,11], including the Czech population [32] and the Danish population [33]. The variant was also reported in the Polish population as associated with the neovascular AMD development [11]. In addition, the A69S variant did not show a significant association with the response to antioxidant supplementation in dry AMD cases, but was involved in worse outcomes of anti-VEGF treatment in neovascular AMD cases [34]. Furthermore, Shijo et al. showed as well that rs10490924was preferably included in PRS and was significant in predicting the need for IAI retreatment and the number of required injections in neovascular AMD patients [35]. The second variant in *ARMS2* reported for Polish population, rs2736911, was consistently included in the predictive model. This association was not confirmed by Wang et al. [36] in non-Hispanic whites. Moreover, the evidence did not show the R38X truncating variant influence on protein levels in retinas [36,37]; however, they retained the protective effect with regard to developing AMD. The last variant incorporated in the model was not previously reported and does not show a splicing effect according to the varSEAK Online Splice Site Prediction tool.


*C2*


Rs9332736 is a 28-bp deletion covering a splice site (donor) causing a complete deletion of exon 6 and a premature stop codon in the Complement Component 2 (*C2*) transcript [38]. It is associated with common type I C2 deficiency and certain autoimmune diseases (e.g., systemic lupus erythematosus), but its effect on AMD has not been reported to date, although the *C2* itself is connected to the AMD phenotype. The deletion is classified as pathogenic/likely pathogenic and the association with AMD should be investigated.


*C3*


Among three variants in the Complement Component 3 (*C3*) gene, rs2230199 was shown in case/control studies to have an impact on AMD in the English, Caucasian and Scottish populations [39,40]. It was confirmed in later genome-wide association studies on the AMD occurrence and progression in the European population [1,12]. As reported by Yan and Seddon, rs2230199 is strongly connected with a faster progression of the disease [12,41], therefore it was included in a multivariant risk model by Seddon. In contrast, rs2230199 did not show statistical significance after Bonferroni correction on MNV in the Czech population [32]. Rs2230199 is associated with changed systemic levels of the components C3d, C5a and C3d/C3 ratios, resulting in altered complement system activation [42], which may reflect in eye complement activity. Studies by Mouallem-Beziere et al. indicated that homozygous GG in rs2230199 resulted in poorer anti-VEGF therapy in MNV patients with large vascularized pigment epithelial detachment [43]. Another variant in the *C3* locus, rs147859257, was extensively investigated by Zhan et al., with the conclusion that the substitution Lys155Gln may interrupt factor H binding to C3b, thus inhibiting C3 protein regulation [44]. Another study showed that resistance to proteolytic inactivation involves not only *CFH* but also *CFI*, leading to the activating alternative complement pathway in AMD pathogenesis [41,45]. The significant association of the rs147859257 was confirmed in those studies on European ancestry populations. The last variant incorporated in the model, chr19:6679578G>A, has not been previously reported and does not show any splicing effect according to the varSEAK Online Splice Site Prediction tool.


*CFB*


The L9H variant was reported as significantly decreasing the risk of AMD in studies on several populations, including Caucasians [34]. Studies show an association of rs4151667 with changed C3d/C3, Ba and Fb complement component systemic levels. Moreover, Gourgouli et al. proved that patients with heterozygous TA and homozygous AA showed a good response to antioxidant treatment and stabilized visual acuity [34]. However, no significant association was obtained for treatment outcomes for neovascular AMD patients. Another two variants, rs4151670 and rs2072634, were reported to the ClinVar database in predisposition screening performed by the Illumina Clinical Services Laboratory (ICSL), among others, as likely benign variants regarding macular degeneration and C2 component deficiency, but no studies on the subject in the European population have been published to this point. 


*CFH*


There are six variants in the complement factor H (*CFH*) gene, including two synonymous and two nonsynonymous substitutions and two intron variants, one of which has not been reported. All previously reported variants are associated with AMD (OMIM) and basal laminar drusen. According to the literature, they have a benign or a likely benign effect on the phenotype. The missense V62I variant and synonymous A473A variant were described by Hageman et al. as being strongly associated with the risk of developing AMD in two analysed cohorts [46]. The substitution of Val-62 with Ile changes the complement component of the C3b binding site, modifying complement pathways and systemic levels of Ba, C3d and C3d/C3 ratio with a potentially protective impact in AMD [28,46,47]. As reported by Shijo and Cobos, rs800292 in PRS may also be significant for predicting the response to intravitreal injection of aflibercept or ranibizumab for exudative AMD [35,48]. The variant rs2274700 was described by Li et al. as strongly associated with AMD in Americans of primarily Western European ancestry [49], while Liao et al. noticed a stronger impact of that variant in Caucasians than in Asian populations, although they avoided conclusions with regard to this association [50]. Additionally, Lee et al. suggested the possible influence of the A473A variant on the vascularized pigment epithelial detachment (vPED) due to AMD [51]. Surprisingly, the best known *CFH* variant with a strong reported association to AMD (rs1061170) was not included in the model and was rejected at the clumping step on behalf of other variants in the *CFH* locus with a higher impact on AMD in the analysed cohorts (possibly on behalf of rs2274700, as Maller, Li and Francis confirmed the stronger association of that variant to AMD [49,52,53]). The intronic variant rs35814900 was identified in predisposition screening performed by the ICSL and up to the date of submission to the ClinVar database it had not been curated or published, and does not show splicing effect as checked in the varSEAK Online Splice Site Prediction tool. Another variant in *CFH*, rs35292876, was included by Fritsche et al. in their 52-variant model on disease progression [1]. Based on these results, Cipriani et al. tested the association of rs35292876 as a single variant and in 8-variant haplotypes with elevated levels of factor H-related protein 4 (FHR-4) in blood, and showed that in the case of a haplotype with altered T the risk of AMD was increased [54]. Their extensive study suggested that FHR-4 may play a prominent role in complement dysregulation in AMD by competing with FH for component C3b binding, while accumulating in the choriocapillaris, Bruch’s membrane and drusen. The next *CFH* variant, N1050Y, was proved by Seddon and Rosner to be statistically significant, showing a protective effect on advanced AMD and GA, as well as for progression to MNV and GA in a univariate analysis. It also has a suggestive protective influence on progression to advanced AMD in one of the multivariate models [55]. However, it was not included in the 13-variant risk prediction model based on regression methods for progression to advanced AMD, GA, or NV, which achieved impressive results of discriminatory ability. The last variant in CFH, chr1:196658497G>T, has not been previously reported to the dbSNP and ClinVar databases, although it has the highest score in the varSEAK Online Splice Site Prediction tool, with an indication to be a splice acceptor variant.


*HTRA1*


The synonymous rs17624021 and intron variant rs2272599 were reported to the ClinVar database in predisposition screening performed by the ICSL laboratory, among others, as benign to macular degeneration, however, no studies on the subject in the European population have been published to this point. The last intronic variant, rs2293871, was evaluated by DeAngelis et al., but the analysis showed a stronger decreasing-risk association of other variants located nearby in the *HTRA1* gene [56].


*NELFE*


The rs522162 in the 3′ UTR region of *NELFE* gene is also located downstream of the *CFB* and *C2* genes, and was reported by Naj et al. as a significantly AMD-risk associated variant [57]. Additionally, Naj analysed gene smoking interactions and found that the variant’s protective association is stronger among ever-smokers than in non-smokers [58]. 


*TLR4*


The TLR4 variant is located in the 3′ UTR region and has not been previously reported to the dbSNP and ClinVar databases.

The established polygenic risk model explains 18% of the variation in age-related macular degeneration in the best outcome. When environmental and demographic covariates are included, the full model reaches 27.5% of the variation explained. Fritsche et al. presented a model based on the effect size weighted sum of identified AMD risk alleles describing 27.2% of disease variability. They evaluated the prediction score based on whole genome data and a larger number of individuals, and therefore the model was calculated on a larger number of variants, including many common polymorphisms located in intergenic and intronic regions. Three variants identified by Fritsche et al. (*C3*: rs147859257, *C3*: rs2230199, *CFH*: rs35292876) were common for our and Fritsche’s risk models [1]. When compared to the 13-variant risk prediction model for progression to advanced AMD, GA, and NV by Seddon [55], we found that three common variants (*ARMS2*: rs10490924, *C3*: rs147859257, *C3*: rs2230199), two of which are in the *C3* gene, were also consistent with the Fritsche model. This implicates the importance of complement component 3 gene variants in both developing AMD and its progression.

Although prediction models with intermediate predictive ability should be interpreted with caution, it is believed that they may be of benefit when utilized in screening programmes for risk prediction, diagnosis support, medical treatment regimens and prognosis [30]. It is worth noting that the most advantageous predictive genetic testing should use both the analysis of rare, highly penetrant variants in AMD-related genes and the effect size weighted impact of common variants on the disease risk [59]. For that reason, exploiting targeted sequencing of AMD-related gene coding sequences together with the presented PRS model may be an affordable solution in AMD risk assessment.

## 5. Conclusions

To the best of our knowledge, this is the first attempt to calculate the AMD risk model for the Polish population, giving the opportunity to expand the analysis of the genetic data to investigate identified variants together with the type of age-related macular degeneration, disease progression and treatment options. 

A possible use of the described model to determine the frequency of ophthalmic visits or qualification for home self-monitoring [5] requires further research, but the results indicate that people aged > 50 years should undergo regular screening if their polygenic risk score is higher than 0.40 (10th decile), as the study shows that almost 92% of those individuals develop AMD.

Making the patient aware of the high probability of AMD can lead to health-related behaviours, such as quitting smoking, which should be advised before the first signs of AMD appear.

## Figures and Tables

**Figure 1 jcm-12-00295-f001:**
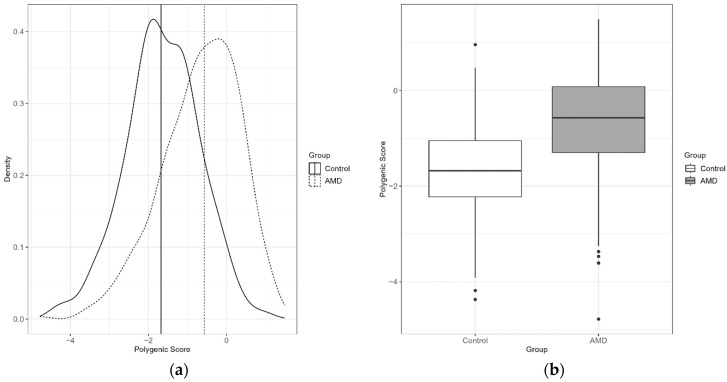
(**a**) Distribution of PRS values per individual as density plot with median values for case/control groups marked as a vertical lines; (**b**) Boxplots showing the distribution of PRS in control and AMD groups.

**Figure 2 jcm-12-00295-f002:**
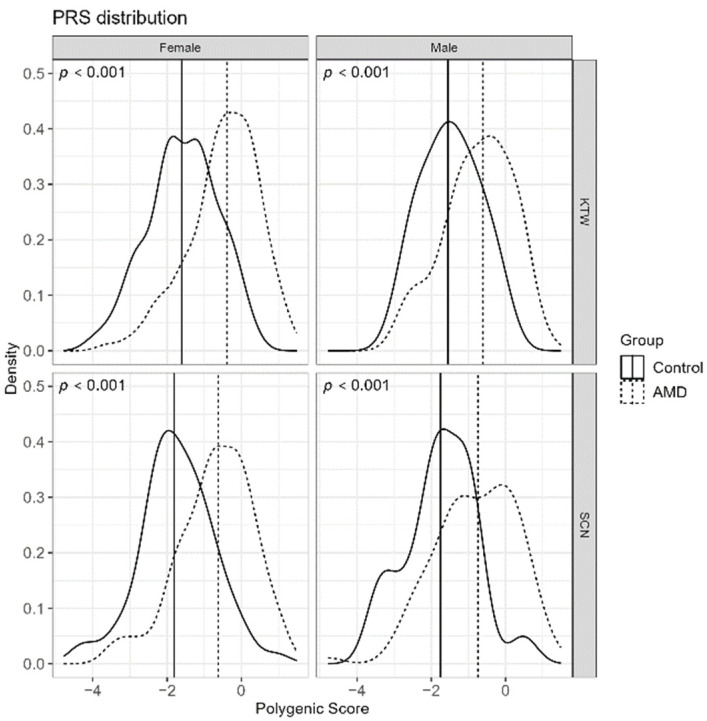
Distribution of PRS values per individual as density plot with median values for case/control groups marked as a vertical line in subsets when sex and the patient recruitment region (Katowice, KTW or Szczecin, SCN) were concerned.

**Figure 3 jcm-12-00295-f003:**
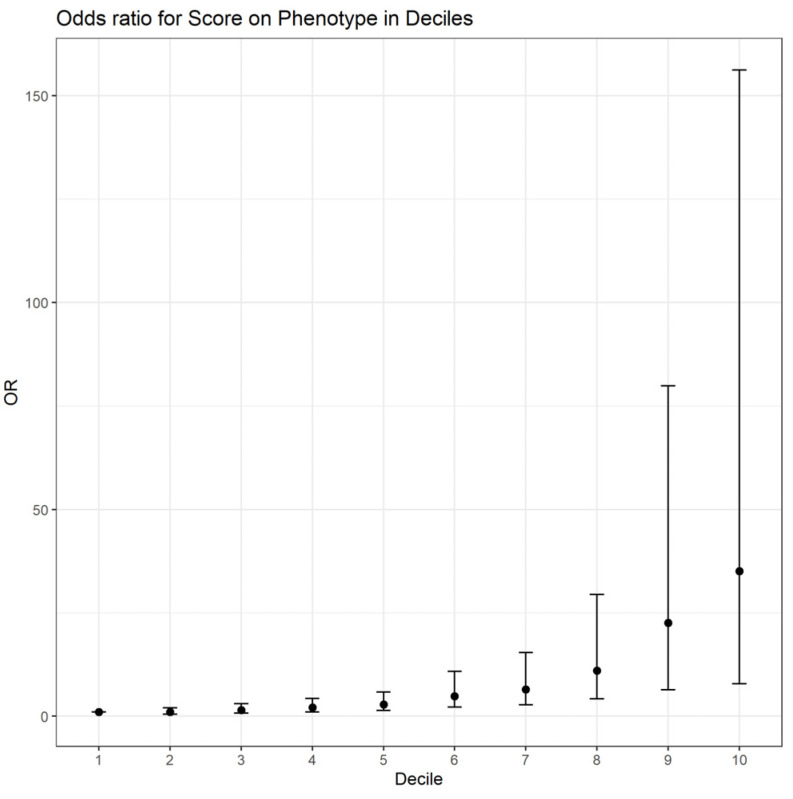
Quantile plot of relative risk of developing AMD. The lowest quantile was set as reference with OR = 1.

**Figure 4 jcm-12-00295-f004:**
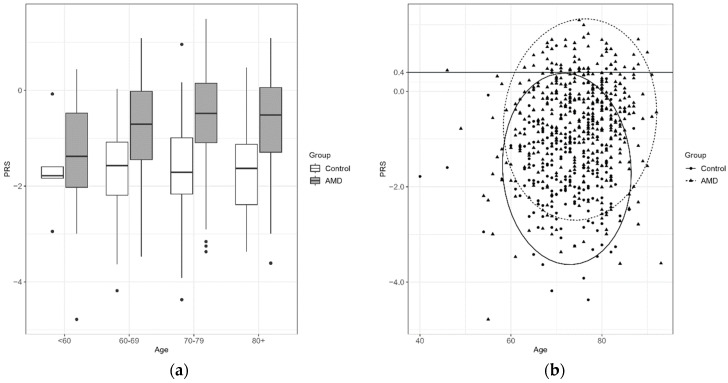
PRS by age (**a**) Boxplot showing the differences in mean PRS in age groups for controls and AMD cases; (**b**) Values of PRS depending on age for controls and AMD cases with 95% confidence level ellipse. The horizontal line corresponds to the 10th decile cut-off at 0.4.

**Table 1 jcm-12-00295-t001:** List of genes included in the enrichment panel.

*ABCA4*	*ARMS2*	*BEST1*	*C1QTNF5*	*C2*	*C3*
*CFB*	*CFH*	*COL8A2*	*EFEMP1*	*ELOVL4*	*ERCC6*
*FBLN5*	*FSCN2*	*GSN*	*GUCA1B*	*HMCN1*	*HTRA1*
*IMPG1*	*PIKFYVE*	*PROM1*	*PRPH2*	*RAX2*	*RP1L1*
*TCF4*	*TGFBI*	*TIMP3*	*TLR3*	*TLR4*	*UBIAD1*

**Table 2 jcm-12-00295-t002:** Characteristics of participants of the age-related macular degeneration study. (*) Mann-Whitney test/Chi-squared test.

	AMD Cases	Controls	*p*-Value *
*N*	471	167	
Enrolment age:			
Mean (SD), years	74.7 (7.9)	72.1 (7.0)	4.3 × 10^−5^
Range, years	[46–93]	[40–87]	
Sex, *N* male/*N* female (%/%)	177/294 (37.6/62.4)	45/122 (26.9/73.1)	0.017
BMI (SD), kg m^−2^	27.2 (4.1)	27.8 (4.7)	0.37
Smoking status:			
Ever smoked, *N* (%)	229 (48.6)	69 (41.3)	0.12
Recent smoker, *N* (%)	73 (15.4)	18 (10.8)	0.18
Pack years, mean (SD)	12.7 (18.4)	8.3 (14.7)	0.005

**Table 3 jcm-12-00295-t003:** Comparison of tested prediction 22-variant models including PRS and (i) the first three genetic multi-dimensional scaling components, (ii) sex, age and smoking status in terms of pack years or (iii) both covariate sets combined.

Model	PRS R^2^	Full R^2^	Null R^2^	*p*-Value
AMD ~ PRS + MDS C1-C3	0.2039	0.2285	0.0247	1.89 × 10^−18^
AMD ~ PRS + sex + age + pack years	0.1974	0.2719	0.0745	8.73 × 10^−19^
AMD ~ PRS + MDS C1-C3 + sex + age + pack years	0.1802	0.2754	0.0951	2.24 × 10^−17^

**Table 4 jcm-12-00295-t004:** Variants with a *p*-value lower than threshold P_T_ = 0.0123 included in PRS model. (†) variants previously reported in studies on AMD. (?) unknown effect on protein, according to HGVS nomenclature.

Gene SNP ID	HGVS Genomic, GRCh37.13	HGVS Coding Amino Acid Change	Effect Allele	Type	EUR Allele Frequency
*ABCA4* rs2297634	NC_000001.10: g.94576968T>C	NM_000350.2: c.302+26A>G p.?	C	intron variant	0.48
*ARMS2* rs2736911 ^†^	NC_000010.10: g.124214355C>T	NM_001099667.3: c.112C>T p.Arg38Ter	T	stop gained	0.14
*ARMS2* rs10490924 ^†^	NC_000010.10: g.124214448G>T	NM_001099667.3: c.205G>T p.Ala69Ser	T	missense variant	0.24
*ARMS2* chr10: 124216397T>C	NC_000010.10: g.124216397T>C	NM_001099667.3: c.298–26T>C p.?	C	intron variant	n.a.
*C2* rs9332736	NC_000006.11: g.31902068_31902095del	NM_000063.4: c.841_849+19del p.Val281fs	A	splice donor, frameshift variant	0.01
*C3* rs147859257 ^†^	NC_000019.9: g.6718146T>G	NM_000064.4: c.463A>C p.Lys155Gln	T	missense variant	0.006
*C3* chr19: 6679578G>A	NC_000019.9: g.6679578G>A	NM_000064.4: c.4457–71C>T p.?	T	intron variant	n.a
*C3* rs2230199 ^†^	NC_000019.9: g.6718387G>C	NM_000064.4: c.304C>G p.Arg102Gly	C	missense variant	0.22
*CFB* rs4151667 ^†^	NC_000006.11: g.31914024T>A	NM_001710.6: c.26T>A p.Leu9His	A	missense variant	0.04
*CFB* rs4151670 ^†^	NC_000006.11: g.31915532C>T	NM_001710.6: c.672C>T p.Tyr224=	T	synonymous variant	0.02
*CFB* rs2072634 ^†^	NC_000006.11: g.31917291C>T	NM_001710.6: c.1365C>T p.Val455=	T	synonymous variant	0.02
*CFH* rs800292 ^†^	NC_000001.10: g.196642233G>A	NM_000186.4: c.184G>A p.Val62Ile	A	missense variant	0.22
*CFH* rs35814900 ^†^	NC_000001.10: g.196642980G>A	NM_000186.4: c.245–7G>A p.?	A	intron variant	0.02
*CFH* chr1: 196658497G>T	NC_000001.10: g.196658497G>T	NM_000186.4: c.965–53G>T p.?	T	intron variant	n.a.
*CFH* rs2274700 ^†^	NC_000001.10: g.196682947G>A	NM_000186.4: c.1419G>A p.Ala473=	A	synonymous variant	0.36
*CFH* rs35292876 ^†^	NC_000001.10: g.196706642C>T	NM_000186.4: c.2634C>T p.His878=	T	synonymous variant	0.01
*CFH* rs35274867 ^†^	NC_000001.10: g.196712596A>T	NM_000186.4: c.3148A>T p.Asn1050Tyr	T	missense variant	0.02
*HTRA1* rs17624021 ^†^	NC_000010.10: g.124249118C>T	NM_002775.5: c.753C>T p.Ile251=	T	synonymous variant	0.05
*HTRA1* rs2272599^†^	NC_000010.10: g.124271595G>A	NM_002775.5: c.1274+14G>A p.?	A	intron variant	0.61
*HTRA1* rs2293871	NC_000010.10: g.124273671C>T	NM_002775.4: c.1275–36C>T p.?	T	intron variant	0.17
*NELFE* rs522162	NC_000006.11: g.31919917T>C	NM_002904.6: c.*161A>G p.?	C	3 Prime UTR Variant	0.09
*TLR4* chr9: 120479716A>G	NC_000009.11: g.120479716A>G	NM_138554.5: c.*2790A>G p.?	G	3 Prime UTR Variant	n.a

## Data Availability

Summary statistics of AMD GWA studies were downloaded from the NHGRI-EBI GWAS Catalog (GWAS Catalog (http://ebi.ac.uk, accessed on 19 October 2022), study accession GCST90086108). The target data can be obtained upon a request from the NeuStemGen STRATEGMED consortium members.

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
