# Peer review of "Polygenic Risk Score Impact on Susceptibility to Age-Related Macular Degeneration in Polish Patients"

_jcm, 2022, doi:10.3390/jcm12010295_

Round 1
Reviewer 1 Report
This paper is well-written, and the literature review is thorough.
1. Why did the authors not perform fluorescein angiography and/or indocyanine green angiography to diagnose AMD?
2. Did this study find which genes could have the greatest impact on AMD progression?
3. Have the authors examined which genes are most strongly affected in subtypes of AMD such as polypoidal choroidal vasculopathy?
Author Response
- Standard methods of ophthalmological imaging were used in the diagnosis - fundus photo, optical coherence tomography and optical coherence tomography angiography. Angiographic studies with contrast (both fluorescein and indocyanin) are invasive and have been performed extremely rarely because they are usually not needed.
- The study focused on AMD development and did not evaluate the impact on specific genes on the disease progression, as there were no follow-up data at the time of preparing the manuscript.
- In the Polish population, the occurrence of PCV is relatively rare and has not been distinguished from the whole group. A deep phenotyping publication is planned in the next step and will include fine subdivisions for detailed retinal morphology.
Reviewer 2 Report
Thank you for the opportunity to review it.
I read it with great interest.
#1 First of all, it seems that the process of determining the 30 types based on what criteria should be properly described in the methods for this genes pack.
#2 Clarify the novelty of this research compared to past research.
#3 It is interesting that genetic mutations are correlated with age and scores are correlated, but isn't it necessary to consider confounding factors in the discussion?
Author Response
- The set of thirty genes involved in the panel have been selected based on Genetics Home Reference (GHR) list of genes associated with AMD (https://ghr.nlm.nih.gov/ condition/age-related-macular-degeneration) at a time of designing the NeuStemGen STRATEGMED project (9 Oct 2015), as described in: Ulańczyk Z, Grabowicz A, Mozolewska-Piotrowska K, et al. Genetic factors associated with age-related macular degeneration: identification of a novel PRPH2 single nucleotide polymorphism associated with increased risk of the disease. Acta Ophthalmol. 2020;99(7):739-749. doi:10.1111/aos.14721
- The past research on the dataset involved analysis of the influence of single variants on the AMD phenotype. The present study estimates the polygenic score of multiple common variants, each of them with a weak effect on the phenotype (development of AMD).
- Age itself determines the disease onset. The high PRS values for younger patients will be evaluated in further studies in terms of correlation with the disease early onset, phenotype severity and type of AMD.